# Innovative Program to Prevent Pediatric Chronic Postsurgical Pain: Patient Partner Feedback on Intervention Development

**DOI:** 10.3390/healthcare12030360

**Published:** 2024-01-31

**Authors:** Danielle Ruskin, Klaudia Szczech, Jennifer Tyrrell, Lisa Isaac

**Affiliations:** 1Department of Psychology, The Hospital for Sick Children, Toronto, ON M5G 1X8, Canada; klaudia.szczech@sickkids.ca; 2Department of Anesthesia and Pain Medicine, The Hospital for Sick Children, Toronto, ON M5G 1X8, Canada; jennifer.tyrrell@sickkids.ca (J.T.); lisa.isaac@sickkids.ca (L.I.); 3Department of Psychology, York University, Toronto, ON M3J 1P3, Canada; 4Lawrence S. Bloomberg Faculty of Nursing, University of Toronto, Toronto, ON M5T 1P8, Canada; 5Department of Anesthesiology and Pain Medicine, Temerty Faculty of Medicine, University of Toronto, Toronto, ON M5S 1A8, Canada

**Keywords:** perioperative psychological program, pediatric chronic postsurgical pain, pediatric transitional pain, patient partners, non-pharmacological pain management, psychological interventions, psychological risk factors, transitional pain service, qualitative analysis

## Abstract

Background: The risk of developing chronic postsurgical pain (CPSP) in youth is related to psychological factors, including preoperative anxiety, depression, patient/caregiver pain catastrophizing, and poor self-efficacy in managing pain. While interventions exist to address these factors, they are generally brief and educational in nature. The current paper details patient partner feedback on the development of a psychologist-delivered perioperative psychological program (PPP) designed to identify and target psychological risk factors for CPSP and improve self-efficacy in managing pain. Methods: Qualitative interviews were conducted with two patients and their caregivers to discuss their surgical and pain management experience and to advise on components of the PPP. Results: Reflexive thematic analysis of interviews generated the following themes, which were incorporated into the content and implementation of the PPP: caregiver involvement, psychological and physical strategies for pain management, biopsychosocial pain education, intervention structure, and supporting materials. Conclusions: The development of a novel psychologist-led PPP is a promising approach to mitigate mental health risks associated with pediatric CPSP and potentially boost postoperative outcomes and family wellbeing. Integrating patient partner feedback ensures that the PPP is relevant, acceptable, and aligned with the needs and preferences of the patients it is designed to serve.

## 1. Introduction

Chronic postsurgical pain (CPSP) is characterized by pain persisting for at least three months or beyond the expected healing time. It is of new onset or has different characteristics or increased intensity from preoperative pain [1]. CPSP is localized to the surgical area or a referred region, and other potential sources of the pain, such as infection or recurrence of injury, have been ruled out. The transition from acute to chronic pain affects an estimated 20–50% of children after major surgeries [2] and is associated with continued analgesic use, longer recovery periods, poorer physical functioning and psychological well-being, missed school days, and increased economic costs [2,3,4,5]. Importantly, youth with recurrent pain experience significant functional impairment and reduced quality of life, including difficulties with peer relationships, disrupted family life, and reduced participation in enjoyable activities [6].

The risk for developing CPSP is highly associated with mental health status, including preoperative depression, anxiety, patient and/or caregiver pain catastrophizing, functional disability, and poor self-efficacy in managing pain, along with high pre- and postoperative pain intensity [2,3]. Furthermore, negative affect and child pain catastrophizing confer additional risk for excessive opioid use postoperatively [7,8,9].

Building upon the impact of CPSP, it is imperative to highlight the significance of psychological interventions in addressing mental health risk factors. Perioperative psychological interventions are designed to reduce anxiety and fear, improve coping skills, and enhance overall wellbeing surrounding the surgical procedure. In pediatric populations, such programs aim to ensure a more positive surgical experience for the patient and their caregivers, as well as improving postsurgical outcomes such as analgesic use, length of hospital stay, and pain intensity [10,11]. Techniques employed include cognitive behavioral therapy strategies (i.e., distraction, guided imagery, relaxation, and coping skills) [12], biopsychosocial pain education [13], therapeutic play [14], surgery demonstrations, and/or tours of the operating room [15]. Preoperative preparation is typically provided by a multidisciplinary team of healthcare professionals including nurses, child life specialists, pediatric psychologists, social workers, anesthesiologists, and surgeons [13,16,17]. Additionally, there has been an increase in digital health interventions using virtual reality [18], web-based [19,20], and mobile applications [21] to deliver psychological interventions to pediatric surgical patients and their caregivers.

To date, surgical preparation and/or preparatory interventions are generally provided by pediatric nursing staff [22,23] and are primarily educational in nature, including procedure-specific information such as the equipment used [24,25], therapeutic role-play [26,27], surgical ward tours [15,28], and distraction strategies [19,29]. Preparatory content is often provided to both patients and caregivers [28,30]. The duration and frequency of guided preparation are highly variable, ranging from 2 weeks before surgery [15] to the day of hospital admission [16]. Most preparation programs consist of single sessions exclusively delivered just prior to surgery [23,31], with few reports of postsurgical follow-up sessions [13].

Although recent systematic reviews [32,33] and a meta-analysis [33] have identified that psychological interventions may be effective in improving acute pre- and postsurgical anxiety, pain, and behaviors in pediatric patients and their caregivers, little data exist on the impact of such interventions on the development of CPSP, persistent opioid use, and other long-term outcomes. To address this gap in the literature, we developed a pilot perioperative psychological program (PPP) incorporating biopsychosocial pain education and non-pharmaceutical psychological strategies to (1) identify and target psychological risk factors for the development of CPSP and (2) improve patient and caregiver self-efficacy in the management of pain throughout the operative period. The program was conceptualized in collaboration with surgeons and healthcare professionals specialized in treating pediatric pain and in partnership with a pediatric Transitional Pain Service (pTPS).

The pTPS was developed to identify patients at risk for CPSP, including those with preexisting pain medication use or experiencing significant impact of pain on function and/or psychosocial factors, irrespective of the type of surgery. The pTPS facilitates the continuity of care between acute and chronic pain services, promotes opioid stewardship, and provides comprehensive multi-modal pain management to decrease the incidence or severity of CPSP. This includes monitoring of opioid and adjunct medication use after hospital discharge, providing support for patients and caregivers in the postoperative period, and ensuring that alternative pain therapies are available [34]. The pTPS is primarily staffed by a nurse practitioner and pain physician, with occasional support provided by psychology, occupational, and physiotherapy as needed, though these professions are not currently part of the core pTPS team.

The pilot PPP is distinct from the pTPS in that each patient and their caregiver receiving the PPP has a dedicated psychology consultation to assess any underlying mental health diagnoses and to identify and attend to risk factors (e.g., pain catastrophizing, anxiety, and low confidence in managing pain) known to place youth at risk for CPSP. Patients and caregivers then learn and practice specific psychological techniques and non-pharmacological strategies with a clinician to manage distress and pain across multiple sessions before and after surgery, while current pTPS care consists of, on average, one appointment before surgery and two postoperative appointments.

In addition to clinical partnerships, successful healthcare program development and implementation requires the engagement of patient stakeholders. Patient partners, defined as individuals (the patients themselves, their caregivers and family members) with lived experience of a health issue who actively collaborate in the research process, have emerged as key stakeholders in pediatric chronic pain research [35]. As outlined in Fox et al.’s [36] recent systematic review on patient partner contributions to health research, “inclusion of patient partners in research is research carried out ‘with’ patients and not ‘on’, ‘about’ or ‘for’ them”. The inclusion of patient partners in research and program design can enhance the relevance and applicability of research findings, foster greater understanding of the experiences lived by those with the conditions under study, and ultimately contribute to improved health outcomes [37]. As a result, calls are increasing for the inclusion of patient partners in health research [38,39,40]. Thus, two former patients who underwent surgery and received pTPS care and their caregivers were invited to participate as patient partners to advise on the pilot PPP in preparation for a randomized controlled study to assess the effectiveness of the intervention. The aim of this paper is to report on the patient and caregiver partner feedback regarding the PPP and how this feedback was incorporated into the program’s development.

## 2. Materials and Methods

### 2.1. Setting

This paper is part of a larger study investigating the feasibility, acceptability, and effectiveness of a pain psychologist-led PPP targeting mental health risk factors for pediatric CPSP delivered via virtual care (PHIPA [Personal Health Information Protection Act]-compliant Zoom for healthcare). This study is being conducted at a pediatric transitional pain clinic providing perioperative preparation and pain management within a large urban tertiary care hospital. Ethics approval for this research study was obtained from the institution’s research ethics board (REB #1000075212).

### 2.2. Criteria for Selection of Patient Partners

In line with the increasing emphasis on the inclusion of patient perspectives in the design and implementation of research programs [41], patient partner feedback on the intervention was sought. Clinical members of the pTPS team were asked to identify two patients who had been seen by the pTPS program and might be suitable to advise on the intervention, with the following criteria: (1) experienced a surgery; (2) received pTPS care either before or after surgery; (3) able to communicate effectively in English; (4) between 8-18 years old. It was determined a priori that two patient partners and their caregivers would provide sufficient feedback based on the resource availability and existing patient partner literature [36]. The selection criteria were chosen to match the eligibility criteria for the randomized controlled study evaluating the PPP. The first two patients and their caregivers who were approached agreed to provide interviews.

### 2.3. Patient Partner Interviews

Interview guides were created to generate discussions with patient and caregiver partners about their experience(s) with surgery and pain management, and to advise on the content and structure of the PPP. The interview guide was developed by the authors and was circulated to other members of the clinic for feedback. A semi-structured interview format was used to encourage rich discussion and reflection and to allow for versatile interviewer–interviewee reciprocity within a guided framework addressing the key components of the PPP [42]. Each interview section consisted of the main questions and follow-up prompts that were structured as open-ended and single-faceted to promote spontaneous, in-depth, and unbiased responses [43]. The full interview guide can be found in Appendix A.

Interviews were conducted in August 2021 by KS, a member of the research team trained in semi-structured interviewing who does not interact with patients clinically, thus reducing the possibility of demand characteristics on responses. Interviews were conducted on PHIPA-compliant Zoom and audio-recorded for transcription. Patient partners received a $20 gift card for their participation in one 1 h interview.

### 2.4. Qualitative Interview Analysis

Interviews were transcribed verbatim and reviewed for accuracy. Following transcription, the data were independently coded by KS and DR using a reflexive thematic analysis approach. This approach involved becoming familiar with the data, generating codes and applicable themes, and refining the themes in an iterative and rigorous manner until the final themes were defined [44]. The PPP was then evaluated with respect to the finalized themes, and any changes were incorporated accordingly.

### 2.5. PPP Prior to Patient Partner Feedback

The PPP was first developed based on evidence that psychological interventions can be used to readily screen for and target mental health risk factors for CPSP [45,46]. It was designed to: (1) improve patient and caregiver self-efficacy in managing and understanding biopsychosocial aspects of pain processing, and (2) address psychological risk factors. The program constitutes three 1 h weekly sessions delivered via virtual care by a psychologist specialized in pain assessment and management, and a phone call “booster” 14 days after surgery. The sessions include education on the biopsychosocial model of pain, training in non-pharmacological pain and anxiety reduction strategies, and the creation of a personalized coping plan for managing pain and distressing thoughts and emotions.

Please note a detailed description of the intervention with the inclusion of patient partner feedback is outlined in Results Section 3.2.7.

## 3. Results

### 3.1. Patient Partners

Two patients who met the criteria were identified by pTPS care providers and, along with their caregivers, were invited to advise on the PPP. Both patients received psychological and physiotherapy support as part of their perioperative care, during which some techniques included in the PPP were provided, contributing to their familiarity with the intervention components. Additionally, since both patient partners received standard pTPS care, they could reliably advise whether the PPP lacked any features that were useful in the perioperative period. See Table 1 and Table 2 for the patient partner characteristics and treatment outcome measures prior to receiving pain management care.

### 3.2. Qualitative Results

The qualitative interview analysis revealed six main themes: (1) caregiver involvement; (2) psychological strategies for pain management; (3) physical strategies for pain management; (4) biopsychosocial pain education; (5) program structure; and (6) supporting materials.

#### 3.2.1. Caregiver Involvement

The patient partners discussed the role of caregivers within the PPP, including the ideal frequency of caregiver participation during sessions. While the interviewees commented that caregivers should be informed of the program elements and engaged in supporting their child, it was also suggested that the therapeutic relationship should primarily be between the patient and therapist for the most effective outcomes:

“*I see both the positive and negative effects of parental involvement, right? So, I think it’s important to have parents on board. Obviously, you need to be transparent with what is going on, but I think ultimately, that service needs to be between the child and the therapist, and I think you’ll get more accurate results if that’s the case.*”Caregiver of 13-year-old female

“*There’s some things that I feel like the parents should know about, like helping the child with reducing anxiety or any, what are their worries or what they’re dealing with. But then maybe some deeper things, like why you’re having anxiety or just some things that you don’t like sharing a lot, you only like sharing to the one person that’s in front of you, would be, I feel like that’s just how I would prefer it*”.13-year-old female

#### 3.2.2. Psychological Strategies for Pain Management

Interviewees identified specific psychological strategies that were helpful in managing their postoperative pain and worries surrounding the experience:

“*The [pain] team gave me some skills to work on and strategies to lower my heart rate, I guess, and my breathing. So I felt a lot better, and I felt safe.*”13-year-old female

“*The psychology helped with talking about how I was feeling and during strategies, different strategies such as meditation, that was something good because when I was in pain, that was one way to calm me down*”.16-year-old male

Importantly, one caregiver whose child was referred to the program postoperatively and therefore only received sessions after their surgery noted how beneficial it would have been to have had the pain management and coping strategies prior to the procedure:

“*I think, though, for him, what might have been good from the get go, knowing what this program offered, was maybe whether it’s finding out beforehand what kinds of things would work for him. So whether it would be music, or whether it would be lighting, whether it would be a meditative thought, those types of things, if we had had that, maybe from the very beginning, that might have helped.*”Caregiver of 16-year-old male

#### 3.2.3. Physical Strategies for Pain Management

In addition to the emphasis on psychological coping strategies, one patient partner suggested a more integrated approach with appropriately incorporated physical activities to address both the psychological and physical aspects of their pain:

“*Yeah, I felt like some of the sessions were just repeating what we just talked about, how to cope with it or how you’ve been feeling and things like that. It was like, okay, I know that information. What’s the next step that I can help with the pain that I’m having right now? I feel like a lot of it, they were separate. I felt like maybe if they’re more intertwined with each other, then that would be good, but I felt like they were two different things and maybe even some more, like physical strategies in the psych stuff. Yeah, that would be perfect.*”16-year-old male

#### 3.2.4. Biopsychosocial Pain Education

Patient partners expressed that beginning the intervention with a biopsychosocial educational component helped them grasp the underlying causes and nature of pain before moving forward to learning coping and pain management strategies. This education contributed to a greater understanding of effective pain management:

“*I also found that that was really helpful for me to kind of understand, because at first, I didn’t really understand why I was having this pain, and, the reason why my body does work like that or anything. But with the psychologist, she really explained that sometimes our bodies can do different things or something’s wrong. So then your brain doesn’t work well with the body signals and everything. So I found that that was really helpful to understand, how the body actually works with pain.*”13-year-old female

“*I think that’s a great way to start, just because a lot of people don’t know right, how pain works and how your brain has a big role in that. So I think that’s a great way to begin, just because it’s informative. And I think when there’s some understanding, then that kind of, I don’t know, helps parents maybe understand a bit more too, right?*”Caregiver of 13-year-old female

#### 3.2.5. Intervention Structure

Interviewees commented on the structure of the program, which was designed as three 1 h weekly sessions prior to surgery and a booster session approximately one to two weeks after. Caregivers highlighted that the program structure provided families with enough time to mentally and practically prepare for the procedure and address any concerns that may arise closer to the surgery date:

“*[Three weeks before the surgery] gives the child and the parent a chance to prepare things that maybe they never even thought of. And three weeks is a good amount of time, when you land something on somebody a week ahead, there’s a lot going on in that one week prior to surgery, people getting off work and taking time and getting the other kids where they need to be. So the three weeks is probably a good target just to get folks, just to initiate them to, hey, this is happening in three weeks. How do you feel about it? Is there anything we should know? And then you get to your session two and three, where they’ve had some time to start pondering things, and then they may have some questions for you that you’d be able to address. Sometimes it takes that long for you to, kind of, for something to come. Especially when there’s stress involved*”.Caregiver of 16-year-old male

It was also noted that the booster session following surgery served as a timely reinforcement of the learned strategies and was important in the continuity of psychological support throughout the recovery process:

“*I found the [postoperative booster session] week after was nice because it was a good check in and a good quick reminder about those strategies, because sometimes you lose that, right, when you’re taking the medication, and you’re forgetting to use those strategies alongside the medication.*”Caregiver of 13-year-old female

#### 3.2.6. Supporting Materials

Physical resources to supplement sessions. Patient partners described the importance of having tangible materials to supplement the intervention sessions for convenience and individual preferences, and to enhance the effectiveness of the program:
“*Maybe in the middle of the night when mum and dad are just dozing off next to you, and you don’t want to wake them, they had a tool that they could grab in their little pouch or whatever it is, whether it’s in the pouch, in their head or something nearby.*”Caregiver of 16-year-old male
“*We didn’t get anything physical, everything was on email. I’m somebody that likes to have maybe a paper in front of me or something like that, so maybe that could be good if it was sent to email and something that you can also print out or use or whatever it is, that would have been good as well.*”16-year-old male
Use of audio/video resources. Interviewees also indicated the need for diverse materials throughout the intervention to cater to different learning styles. Visual interactive learning tools such as diagrams and videos could enhance comprehension and provide clear step-by-step guidance, particularly for complex strategies that may be difficult to understand through verbal explanation alone:
“*Yeah, maybe like a diagram. Yeah, because I’m also a visual learner, so, um, sometimes words don’t really, like, go through my brain. I have to look at something, to help me understand.*”13-year-old female
“*I think the videos like, having the videos sometimes, if there was something to kind of exemplify it. I think having, like, a video, maybe explaining, maybe more your having, explaining more of your, you know, more complicated kind of strategy strategies would be helpful. But yeah the video thing. Like, almost kind of instructional videos kind of thing. Yeah. Where it’s almost like a YouTube, like, showing you how to do it kind of thing.*”Caregiver of 13-year-old female

#### 3.2.7. Revisions to the PPP Based on Patient Partner Feedback 

Patient partner feedback was incorporated into finalizing the PPP’s design and implementation. The program will still consist of three 1 h sessions delivered prior to surgery and provided approximately weekly, beginning three weeks before surgery; however, the booster session will now be provided via PHIPA-compliant Zoom for healthcare (as opposed to telephone) 10–14 days following surgery. The booster session will be scheduled after the patient has had their postoperative pTPS appointment to ensure all questions and/or concerns are adequately addressed. Although all attempts will be made to follow the described program structure, the sessions are designed to be flexible and can be combined in instances where there is less than three weeks between the program enrollment and the scheduled surgery date. Similarly, although the program is delivered within a standardized framework, sessions are flexibly designed so that specific content, techniques, and skills may have more or less time and attention based on presenting issues and to avoid unnecessary repetition. Finally, patients and caregivers emphasized the importance of protected time between the patient and clinician to discuss private and/or sensitive information. In response, caregivers will be invited to join at the end of each session.

Regarding the program content, the sessions will include a consultation (clinical interview and standardized questionnaires for patients and caregivers), biopsychosocial education outlining physiological pain mechanisms and the mind-body connection, and an introduction and iterative review of non-pharmaceutical pain management skills (both psychological and physical strategies). The session content was organized according to patient partner feedback (e.g., biopsychosocial education should be delivered in Session 1). Participants will also receive a customized pain treatment plan with their most effective pain management and coping strategies (i.e., medications, physical and psychological strategies, personal items, etc.) designed to assist with recovery after surgery.

Sessions will include activities such as clickable web links to audio/video resources and fill-in-the-blank components to be completed interactively by the patient and clinician during sessions. As an adjunct to the virtual care format, the program content will also be delivered via personalized slides accompanying each session that can be printed as a handout or saved electronically on personal devices. The slides will be sent to the families via email after each session. The slides can then be easily accessed by the families outside of the sessions and used as a resource while in the hospital. See Table 3 for a summary of the themes and changes implemented to the PPP and Table 4 for a detailed description of the PPP currently being investigated in an ongoing pilot randomized controlled trial.

## 4. Discussion

There is a growing need to establish comprehensive perioperative pain management programs incorporating multidisciplinary interventions and collaborative approaches [47], particularly in pediatric populations. Such programs are essential for providing high-quality, patient-centered care that addresses the complex needs of pediatric surgical patients and their families, and their implementation has the potential to improve immediate and long-term surgical outcomes, enhance patient and caregiver satisfaction, and optimize healthcare resources [48]. The development of a pain psychologist-led perioperative psychological program (PPP), as detailed in this manuscript, represents a significant advancement in pediatric perioperative care. The PPP addresses a critical gap in the prevention of chronic postsurgical pain (CPSP) by targeting mental health risk factors that have been shown to increase the risk of poorer postoperative health outcomes, including longer hospital stays, increased pain intensity, and excessive medication use [3].

The PPP provides specific attention to psychological factors that can contribute to the risk of developing CPSP. This intervention adds tailored one-to-one psychological services provisioned by a clinical psychologist specializing in pain assessment and management to an established pediatric Transitional Pain Service (pTPS). Preoperative psychological preparation has been shown to reduce patient and caregiver catastrophizing [49], fear and anxiety [22], and overall pain [13]; however most studies report brief (30–60 min) single-session programs provided at admission or as a preoperative consultation [14,16,31].

The current program’s structure is novel in that virtual sessions are provided beginning several weeks in advance of the surgical procedure, allowing the patients and families to ask questions related to the upcoming procedure with sufficient time for responses from their healthcare team, mentally prepare for the upcoming event, and practice the learned pain/distress management skills so they can be used in the perioperative period. The learned strategies are then reinforced in a postoperative booster session, a strategy suggested to improve pain management in pediatric surgical populations [50]. Relatedly, recognizing that the inclusion of caregivers significantly improves child symptoms for painful conditions immediately post-treatment [51,52], the sessions are designed to incorporate both patient and caregiver participation. Furthermore, the interdisciplinary and collaborative nature of the current program encourages continuity of care in the perioperative period and facilitates the use of various resources such as Child Life and community-based support programs. Finally, the program is designed to be flexible and individualized to the needs of each patient.

Integrating patient partner feedback into the program’s design and implementation ensures the intervention aligns with clinical expertise, evidence-based practice, and the lived experiences of those it aims to serve. Qualitative data gleaned from semi-structured interviews with patient and caregiver partners were instrumental in shaping the content and delivery of the PPP. First, patient partners identified a need for balance between caregiver support and providing a confidential space for the patient and clinician to discuss private and/or sensitive information. In turn, caregivers are invited to join at the end of each session. Next, patient partners felt the structure of the PPP (i.e., three virtual sessions prior to surgery and one after) was appropriate, and suggested that the “booster” session be changed from a phone call to Zoom. With respect to the program content, patient partners discussed specific psychological and physical strategies and techniques that were effective in the perioperative period, and the order in which the content is best delivered. This facilitated the organization of the content in a standardized framework (e.g., biopsychosocial education delivered in the first session with instruction and the practice of pain and distress management skills in later sessions), that can be delivered flexibly depending on the presenting concerns. Finally, patient partner feedback led to the development of a personalized slide deck to be completed interactively with the clinician during sessions. The slide deck includes weblinks to additional resources (i.e., instructional videos and virtual hospital tours and demonstrations) that can be accessed outside of sessions on personal electronic devices or printed as a physical handout. This patient partner-informed approach provided crucial insights that enhanced the intervention, making it more relevant and acceptable to pediatric patients and their families. The ongoing randomized controlled trial investigating the acceptability and effectiveness of the PPP includes eliciting qualitative feedback and satisfaction from PPP participants, which will assist in further refining the intervention.

This study is, to our knowledge, the first to report patient and caregiver partner feedback on the development of a perioperative psychological program targeting the prevention of pediatric CPSP. There are, however, several limitations that need to be considered. First, the small number of interviewees limits the generalizability of our findings. While a recent systematic review reported that 36% of studies engaged between one and five patient partners [36], specific guidelines should be established for the inclusion of patient partners in research (i.e., suggested number of partners, level of engagement, and compensation and/or authorship). Second, although patient partners received elements of the PPP in their perioperative care, their feedback is limited in that they did not participate in the PPP intervention itself. To address this, we are continually collecting feedback as part of the larger ongoing study evaluating the effectiveness of the PPP. Finally, the data collection instrument (i.e., a semi-structured qualitative interview guide) is a limitation as it is a subjective measure developed by the authors for the purposes of this study, with limited capacity to conduct formal statistical analyses. The larger study includes both qualitative feedback interviews and quantitative satisfaction measures to more comprehensively evaluate the PPP from patient, caregiver, and provider perspectives.

## 5. Conclusions

The developed perioperative psychological program represents a promising strategy for addressing mental health risk factors associated with chronic postsurgical pain in pediatric patients. The inclusion of patient and caregiver partners in the program design and implementation represents a comprehensive approach to the provision of perioperative care. The current program was modified according to patient and caregiver partner feedback to include protected time between the patient and clinician with caregiver involvement as appropriate, an introduction to biopsychosocial pain education, psychological and physical techniques to manage pain and distress delivered in a standardized but flexible format, and multimodal materials to supplement intervention sessions and enhance patient engagement. The modified program has the potential to reduce procedural distress and improve confidence in managing pain prior to and after major surgery and to be a demonstration case for the possibility of adding more mental health specialty services to a transitional pain service. More broadly, addressing risk factors prior to surgery is ultimately hoped to reduce the length of stay and healthcare burden associated with the chronification of pain. This tailored and flexible program will continue to be evaluated and refined using patient and caregiver feedback as part of an ongoing randomized controlled trial. In sum, integrating patient-centered methodologies in healthcare research and intervention design is essential to meet the dynamic needs of patients and families, and to ensure that care is delivered effectively and compassionately.

## Figures and Tables

**Table 1 healthcare-12-00360-t001:** Patient partner characteristics.

Patient Partner	Age	Sex	Number of Prior Surgeries	Most Recent Surgery	Number of Appointments ^1^		Caregiver Co-Interviewed
					Prior to Surgery	After Surgery	
1	13	Female	2	Orthopedic hardware removal + nerve repair	8	1	Mother
2	16	Male	1	Umbilical hernia repair	0	11	Mother

^1^ Pain management appointments include treatment as provided by the pTPS and a combination of psychological and physiotherapy support as needed.

**Table 2 healthcare-12-00360-t002:** Patient partner treatment outcome measures prior to receiving pain management care.

Measure	Patient Partner 1	Patient Partner 2
Patient Reported Measures		
Pain NRS—Usual pain in the last week ^1^	5	7
Pain NRS—How bothered/upset when pain is at a “5” ^2^	5	10
PROMIS Pain Interference ^3^	40.6	44.3
PROMIS Mobility ^3^	45.0	43.0
PROMIS Anxiety ^3^	33.5	51.2
PROMIS Depressive Symptoms ^3^	35.2	50.6
Pain Catastrophizing ^3^	41.5	62.0
Caregiver Reported Measures		
PROMIS Pain Interference Parent Proxy ^3^	63.0	46.0
PROMIS Mobility Parent Proxy ^3^	no data	42.0
PROMIS Anxiety Parent Proxy ^3^	64.2	54.7
PROMIS Depressive Symptoms Parent Proxy ^3^	36.2	45.4
Pain Catastrophizing ^3^	58.4	48.3

^1^ Measured on a scale of 0 (no pain) to 10 (pain as bad as you can imagine); ^2^ Measured on a scale of 0 (not bothered at all) to 10 (very bothered); ^3^ Represented as *t*-scores (Mean = 50, SD = 10) with higher scores indicating more of the domain being measured; NRS = Numerical Rating Scale; PROMIS = Patient-Reported Outcomes Measurement Information System; SD = standard deviation.

**Table 3 healthcare-12-00360-t003:** Themes identified in semi-structured interviews and changes incorporated in the PPP.

Theme	Summary of Feedback	How Was Feedback Incorporated?
Caregiver involvement	Patients appreciated the confidential time to disclose private/sensitive information to the clinician but also recommended caregivers be involved so they could undertake a supportive role throughout the intervention.	✓ Patients provided with protected time with the clinician to discuss confidential information. ✓ Caregivers invited to participate at the end of each session. ✓ Session materials (slide decks personalized to each patient) shared with the caregiver at the end of each session.
Psychological strategies for pain management	Psychological skills and strategies to cope with anxiety and pain were identified (e.g., meditation, deep breathing, and methods of distraction).	✓ All sessions include psychological skills/distress management strategies (e.g., belly and box breathing, grounding, cognitive behavioral exercises to challenge worried thoughts, progressive muscle relaxation, and guided imagery).
Physical strategies for pain management	Physical strategies for symptom management were suggested to supplement the psychological skills included in the intervention.	✓ Session 3 includes physical pain management strategies (e.g., cold/hot, stretching, massage, and resting) and guided movement after surgery (e.g., pacing and changing position).
Biopsychosocial pain education	Pain education (i.e., how pain works in the body and the mind-body connection) was identified as an essential intervention component best provided in the first session.	✓ Session 1 introduces the biopsychosocial approach to pain management (i.e., medication and physical and psychological strategies) and pain processing mechanisms (e.g., gate control theory and fight-or-flight response).
Intervention structure	The optimal intervention structure including virtual care delivery, the timing of sessions relative to surgery, and the session duration was outlined.	✓ The PPP is three 1 h weekly sessions starting three weeks before surgery when possible. Booster session 10–14 days after surgery via Zoom. ✓ Sessions designed to be flexible, such that they can be combined into a longer appointment when there is limited time prior to surgery, and more/less time may be spent on specific techniques depending on presenting issues and to avoid repetition.
Supporting materials	Physical resources were recommended to supplement the virtual sessions. Printable copies of the session content were suggested to be especially valuable while in the hospital for surgery. Audiovisual resources were suggested to increase engagement during the sessions and facilitate practice between the sessions.	✓ Printable slides created to accompany each session so patient and clinician could interactively complete session activities. ✓ Completed handouts personalized to the patient can be printed and brought to the hospital. ✓ Session slides include links to audiovisual resources (e.g., instructional videos, YouTube links, hospital resource pages including virtual tours and demonstrations), which can be accessed outside of sessions.

**Table 4 healthcare-12-00360-t004:** Description of finalized PPP for youth undergoing major surgery.

Session	Timing Relative to Surgery	Content
1	3 weeks before	Consultation (30–60 min) Psychologist-led consultation comprising a patient and caregiver clinical interview with standardized questionnaires to identify underlying mental health diagnoses, psychological risk factors for CPSP, and/or worries related to the surgical experience. Education (30 min) Education on the biopsychosocial model of pain and how links between mind and body can affect pain (e.g., gate control theory). Having several different ways to manage pain and reduce distress is better than relying on just one way (e.g., exclusively medication). Introduction to non-pharmacological pain management strategies (e.g., belly breathing, relaxation). Review the session content and set practice goals for the following week. Materials sent to the family via email.
2	2 weeks before	Review of Session 1 content and practice goals. Discuss any concerns surrounding the surgical experience. Target worries and/or risk factors with cognitive behavioral techniques to reduce anxiety and worrying about surgery and pain (e.g., thought recording and guided imagery). Prepare a calming tool kit with practical and individualized non-medicine pain and anxiety reduction strategies that the patient can bring to the hospital (e.g., relaxation techniques, physical strategies, and personal belongings). Review the session content and set practice goals for the following week. Materials sent to the family via email.
3	1 week before	Review of the Session 1 and 2 content and practice goals. Practice learned skills and physical strategies that can help with pain after surgery (e.g., hot/cold, stretching, pacing, and changing position). Develop a personalized coping plan (i.e., medication and physical and psychological strategies) and review the calming tool kit in preparation for surgery. Review the resources, skills, and strategies that can be used before and after surgery. Set the practice goals for the following week. Materials sent to the family via email.
Booster	10–14 days after	Address any challenges or concerns in recovery. Review the most helpful strategies from the personalized coping plan and calming tool kit. Practice learned skills and strategies to help with recovery going forward. Follow-up referrals as applicable (e.g., counseling and pTPS). Set practice goals for the future. Materials sent via email to the family.

## Data Availability

The data presented in this study are available on request from the corresponding author. The data are not publicly available due to Institutional Research Ethics Board restrictions.

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
