# Peer review of "Innovative Program to Prevent Pediatric Chronic Postsurgical Pain: Patient Partner Feedback on Intervention Development"

_healthcare, 2024, doi:10.3390/healthcare12030360_

Round 1

Reviewer 1 Report

Comments and Suggestions for Authors

The fight against pain has been one of the most important problems of mankind for a long time. Various medications used in the postoperative period are certainly important. At the same time, the potential of psychological techniques in this area has not been fully used. Therefore, the research presented by the author is both relevant and in demand. It is necessary to emphasize the special importance of this study in view of the fact that it is conducted in a pediatric clinic. The authors proved that the proposed program contributes to the prevention of chronic postoperative pain.

The authors conducted a large-scale qualitative study. However, for more convincing results, there is not enough at least expert assessments on a metric scale (with/without a program). It would also be desirable to introduce a standardized observation that would help assess the increment in pain, anxiety, stress, etc. Obviously, all this can be taken into account in future studies.

The fact is that we have to choose in this case: what is more important is the evidence or the significance of the problem being solved.  It seems to me that the problem being solved is more significant.  Therefore, I recommended publication without much correction.  Researchers can no longer interview the subjects.  The only thing I would recommend doing in this article is to expand the presentation of the program itself with explanations of what each stage is aimed at.  The following comments can be used in next research.

Despite the fact that the article presents the results of a qualitative analysis of a series of interviews, it can be supplemented with some quantitative measurements based on self-scaling of various characteristics of pain and the impact on its experience as a result of the implementation of the program.  In addition, it would be appropriate to more specifically highlight a number of specific procedures of the program itself (before correction) and provide theoretical justifications for the stages of solving the problem.  Finally, projective diagnostic methods would also be appropriate, which would help in the future to more unambiguously diagnose the localization and experience of pain.  An important point in experiencing pain is the general sensitivity of the nervous system, so it would also be appropriate to make arguments about the applicability of the program for people with different pain thresholds and general sensitivity.

Reviewer 2 Report

Comments and Suggestions for Authors

Ruskin et al. wrote about patient and caregiver feedback on a perioperative psychological support program they developed.  This topic is definitely understudied and important to Healthcare readership.  The paper is well written.  Attention to feedback below will increase it's suitability for publication.  I appreciate the authors' efforts and hope that they find my feedback beneficial.

-It would be helpful to the reader to provide more clarity on how the PPP differs from the pTPS.  Specifically, how does the PPP differ from the psychological and physiotherapy care the patient partners received?  

-It would be helpful to have more information on how the interview guides were created (for example, lit review? Clinical experts?).  It would similarly be helpful to include the interview guide in a table or in supplemental materials, as appropriate.  This provides the reader with a better understanding of the structure for data collection, and how patient partners were able to provide feedback.

-I am concerned about the low N (two patients and two caregivers providing feedback).  Please provide the total number of patients who met eligibility criteria so that the reader can have a better idea of the participation rate.  

-The small N should be listed as a limitation.  Their feedback is further limited because they did not participate in the PPP intervention.  

-I am not entirely clear how the authors incorporated patient partner feedback.  I appreciate outlining changes made in Table 2, but it seems that there is a lot of feedback that went unaddressed when modifying the intervention.  For example, the authors described common techniques for perioperative psychological intervention beginning pg 2 (line 96). Some of these elements were included in their PPP, but others were not.  Were the patient partners asked whether or not surgery demonstration, tours, therapeutic play, etc. would have been helpful to include? Or were they only asked to comment on existing techniques in the PPP? One the of patient partners commented on the repetition of the sessions; how did the authors address this when incorporating feedback into the intervention? Similarly, another patient partner suggested making physical and psychological techniques more intertwined; how was this addressed?  Were patient partners able to provide open-ended feedback at all?  Greater detail on the parameters for soliciting feedback, and how the feedback was and was not incorporated into the intervention, would significantly improve the utility of the paper.

-The discussion section would be improved with narrowed focus on patient partner feedback and how the PPP was modified, rather than the novelty or benefit of the intervention.

-The authors are encouraged to also include literature on single-session perioperative programs (e.g., Darnall) when discussing benefits of multi-sessions. I would be helpful to discuss how their partner feedback played into their decision to make it multi- versus single-session.  

-Please expand on what is meant by, "The integration of the pediatric Transitional Pain Service (pTPS) and patient partner feedback into the program's design and implementation..." (pg. 10, beg line 346). Did clinicians from pTPS also provide feedback on the PPP? Given the small N and limited feedback, it may be helpful to also include this feedback as well.

Reviewer 3 Report

Comments and Suggestions for Authors

Dear Editor

The article "A novel perioperative psychological program for prevention of paediatric chronic postsurgical pain: Description of patient partner feedback on intervention development" deals with a relevant topic in the field of paediatric health.

The article is well-founded and the methodology adopted is clearly described. The results are presented and discussed.

However, a few questions could be made, namely:

Point 1.1. Summary of Current Perioperative Preparation and Psychological Interventions, is out of context, that is, not the content itself but the title does not make sense to be within the introduction. I suggest removing the subheading and using the content to finish the introduction.  Then add the paragraph describing the aim of the article.

The methodology seems to be well explained, but the problem with the article is that it only uses two cases. You could therefore consider changing the title to an exploratory study.

When presenting the results, it might be important to better focus the situation of young people with chronic postsurgical pain. Note that the situations are not similar. One involves minor surgery  and the other is more complex and requires more consultations and medical procedures, so the impact of pain (physical and psychosocial) is not similar. That said, it would be relevant to identify the carers' perspective on the process, after this discussion.    

The discussion of the results could highlight in greater detail the results of the exploratory study carried out in the article, namely what the carers said was relevant to the support they received or needed to understand chronic paediatric pain.

These are some questions that could improve the article presented.

Reviewer 4 Report

Comments and Suggestions for Authors

Dear authors,

I appreciate the opportunity to review the study titled "[ A novel perioperative psychological program for prevention of pediatric chronic postsurgical pain: Description of patient partner feedback on intervention development]." The topic is interesting, and I look forward to contributing to the review process.

The aim of the paper, focusing on patient and caregiver partner feedback regarding the Perioperative Psychological Program (PPP) and its incorporation into the program's development, is noteworthy.

I hope these suggestions enhance the article's clarity, structure, and impact.

Title:

Consider shortening the title for conciseness while maintaining clarity: ex.

Revised Title: "Innovative Program to Prevent Paediatric Chronic Postsurgical Pain: Patient Partner Feedback on Intervention Development"

Introduction:

Expand on CPSP Impact: To offer a more detailed understanding of how CPSP affects the daily lives of paediatric patients and their families, consider including real-life examples or narratives that highlight the challenges faced during the immediate postoperative period and the lasting consequences on patients' well-being. Emphasize the potential long-term impact on physical functioning, psychological well-being, and overall quality of life.

Highlight the Significance of Psychological Interventions:

Elaborate on the crucial role of addressing mental health risk factors in paediatric CPSP. Connect it explicitly to patients' overall well-being and quality of life, emphasizing how psychological interventions can improve surgical outcomes, reduce pain intensity, and enhance recovery. This linkage will underscore the significance of integrating psychological support into the perioperative care of paediatric patients.

Improve Transition Sentences:

Example: To enhance the transition between paragraphs, consider revising sentences like, "Building upon the impact of CPSP, it becomes imperative to highlight the significance of psychological interventions in addressing mental health risk factors." This ensures a smoother flow of ideas and logically connects each point.

Methods:

Clarify Patient Partner Selection Process:

Provide a more in-depth explanation of how pTPS care providers identified patient partners. Include specific criteria or considerations used in the selection process to enhance transparency and provide readers with insights into the rationale behind the selection.

Include Participant Characteristics:

Consider adding a brief description of the demographic characteristics of the patient partners. This addition will provide context for readers, allowing them to understand better the perspectives brought by patient partners to the study.

Results:

Linkage to Program Development:

Explicitly connect each identified theme to the corresponding changes made in the PPP design. This direct linkage will help readers appreciate how patient feedback directly influenced the development of the program, reinforcing the importance of patient perspectives.

General Recommendations:

Consistent Terminology:

Ensure consistent use of terminology throughout the article. For example, choose whether to refer to the intervention as the "proposed PPP" or the "developed PPP" and maintain this consistency throughout the manuscript.

Discussion

The discussion provides a comprehensive overview of the significance and innovative aspects of the proposed Pain Psychologist-led Perioperative Psychological Program (PPP). However, there are areas where the discussion can be improved for clarity and conciseness:

Condense Sentences for Clarity:

Original: "The integration of the pediatric Transitional Pain Service (DPS) and patient partner feedback into the program's design and implementation ensures that the intervention is grounded in clinical expertise and evidence-based practice and resonates with the lived experiences of those it aims to serve."

Suggested: "Integrating the pediatric Transitional Pain Service (pPS) and patient partner feedback ensures the intervention aligns with clinical expertise, evidence-based practice, and the lived experiences of its recipients."

Combine and Streamline Sentences:

Original: "Qualitative data from semi-structured interviews with patient partners (i.e., patients and their caregivers) were instrumental in shaping the PPP's content and delivery. Patient partners provided valuable feedback regarding the balance of caregiver support with the need for a patient to have a private space with their therapist, the value of tailoring psychological and physical pain management strategies alongside educational content and providing multimodal materials to supplement the intervention."

Suggested: "Valuable insights from patient partners, obtained through semi-structured interviews, played a crucial role in shaping the content and delivery of the PPP. Their feedback covered the balance of caregiver support, the importance of a patient's private space during therapy, the value of tailoring psychological and physical pain management strategies, and including multimodal materials to enhance the intervention."

Enhance Transition between Paragraphs:

Original: "The proposed PPP constitutes an innovation in pediatric perioperative care in several ways."

Suggested: "In summary, the proposed PPP represents a groundbreaking innovation in pediatric perioperative care for several reasons."

Clarify Phrasing:

Original: "The ongoing randomized controlled trial investigating the acceptability and effectiveness of the proposed PPP includes eliciting qualitative feedback and satisfaction from PPP participants, which will assist in further refining the intervention."

Suggested: "The ongoing randomized controlled trial, aimed at assessing the acceptability and effectiveness of the proposed PPP, involves gathering qualitative feedback and participant satisfaction. This information will be instrumental in refining the intervention further."

Implementing these suggestions will enhance the discussion’s clarity and make it more concise while retaining its informative content.

Conclusion:

  1. Restate Key Findings:
    • Summarize the key findings from the patient partner feedback and emphasize how these findings shaped the final PPP design.
  2. Future Implications: Please briefly discuss the developed PPP's potential implications, such as its broader application in other healthcare settings or its integration into standard care practices.

  1. Call to Action:End with a call to action, emphasizing the significance of patient-centered approaches in healthcare research and the need for continued improvement and adaptation.

Round 2

Reviewer 2 Report

Comments and Suggestions for Authors

I really appreciate the authors' careful attention to reviewer feedback and feel that their changes have resulted in a much clearer, stronger paper.  There are two things to consider:

-Please include SDs on Table 2 for those measures with a t score

-I understand that the authors wish to highlight the novelty of their work (pg 9, lines 452-475).  I continue to strongly suggest that they reconsider, as focusing on the novelty seems to detract from their primary findings (modifications to their program based on patient partner feedback).  More importantly, there are published programs for peds pre-op pain preparation, so the authors' use of partner feedback is what is truly novel (the authors are encouraged to consider works by Edwards, Ramezani, Rabbitts, and Brimeyer who have published programs or frameworks).  While this type of program might not be novel, their patient partner feedback is and I would suggest it be highlighted.  If the authors choose to keep this paragraph, I would temper the language to include how their work differs from previously published works.     
